# Inhibition of Dermatophyte Fungi by Australian Jarrah Honey

**DOI:** 10.3390/pathogens10020194

**Published:** 2021-02-11

**Authors:** Annabel Guttentag, Krishothman Krishnakumar, Nural Cokcetin, Steven Hainsworth, Elizabeth Harry, Dee Carter

**Affiliations:** 1Marie Bashir Institute, School of Life and Environmental Sciences, University of Sydney, Sydney, NSW 2006, Australia; annabel.guttentag@sydney.edu.au (A.G.); krishoth2@gmail.com (K.K.); 2ithree Institute, University of Technology Sydney, Sydney, NSW 2007, Australia; nural.cokcetin@uts.edu.au (N.C.); elizabeth.harry@uts.edu.au (E.H.); 3School of Science, RMIT University, Melbourne, VIC 3083, Australia; shainsworth1591@gmail.com

**Keywords:** honey, antifungal activity, *Trichophyton rubrum*, dermatophytes, hydrogen peroxide

## Abstract

Superficial dermatophyte infections, commonly known as tineas, are the most prevalent fungal ailment and are increasing in incidence, leading to an interest in alternative treatments. Many floral honeys possess antimicrobial activity due to high sugar, low pH, and the production of hydrogen peroxide (H_2_O_2_) from the activity of the bee-derived enzyme glucose oxidase. Australian jarrah (*Eucalyptus marginata*) honey produces particularly high levels of H_2_O_2_ and has been found to be potently antifungal. This study characterized the activity of jarrah honey on fungal dermatophyte species. Jarrah honey inhibited dermatophytes with minimum inhibitory concentrations (MICs) of 1.5–3.5% (*w*/*v*), which increased to ≥25% (*w*/*v*) when catalase was added. Microscopic analysis found jarrah honey inhibited the germination of *Trichophyton rubrum* conidia and scanning electron microscopy of mature *T. rubrum* hyphae after honey treatment revealed bulging and collapsed regions. When treated hyphae were stained using REDOX fluorophores these did not detect any internal oxidative stress, suggesting jarrah honey acts largely on the hyphal surface. Although H_2_O_2_ appears critical for the antifungal activity of jarrah honey and its action on fungal cells, these effects persisted when H_2_O_2_ was eliminated and could not be replicated using synthetic honey spiked with H_2_O_2_, indicating jarrah honey contains agents that augment antifungal activity.

## 1. Introduction

Dermatophytes are filamentous fungi that cause superficial infections of the hair, skin, and nails. These infections, collectively described as tineas, are some of the most prevalent fungal ailments, and it has been estimated that almost everyone will acquire an infection at some point of their lifetime [1]. Over the last few decades, the incidence of tinea of the nails (onychomycosis) has risen from 2% to 14% in the developed world [2], and tinea pedis or athletes foot has increased to around 20–25% of all adults [3]. Many factors have been implicated in this rising incidence, including increasing travel, pet ownership, and sporting facility use, along with an aging population [3]. The long duration of many dermatophyte infections presents a significant morbidity burden, especially for vulnerable groups like athletes and military personnel [4].

Currently, around 500 million USD is spent annually worldwide on antifungal therapies for tineas [1]. Treatment can be either systemic or topical, with the latter favoured due to easier self-administration and less severe side-effects [5,6]. However, these therapies often take weeks to months with daily or twice-daily applications, and relapse is common [6]. These limitations have stimulated interest in novel tinea therapies, including the use of natural products.

Honey has been used in cosmetics and medicines as an antimicrobial, emollient, and humectant since ancient times, and is still used extensively in a variety of modern cosmetics, with some recent licensing of sterilized honey for clinical use [7]. The antimicrobial effects of honey have been reported for a wide range of fungi [8] and bacteria [9,10,11]. Honey is a supersaturated sugar solution of glucose and fructose sugars, and it also contains a diverse range of proteins of bee and plant origin, as well as pigmented and antioxidant compounds including polyphenols, flavonoids, and Maillard reaction products [12,13,14,15]. Various components of honey give it antimicrobial properties, including high sugar and low pH. The most significant antimicrobial activity is attributed to either the production of hydrogen peroxide (H_2_O_2_) or to “non-peroxide” nectar-derived chemicals. H_2_O_2_, a powerful oxidant, is produced by glucose oxidase, a bee-derived enzyme that is activated upon dilution of honey with water [15]. “Non-peroxide” honeys generally originate from manuka plants of the *Leptospermum* genus and contain methylglyoxal (MGO), a toxic compound that crosslinks and inhibits microbial proteins [16,17]. Additional non-peroxide floral factors include phenolic compounds and peptides. The standard methodology to differentiate between these two types of antimicrobial activity is to treat honey with a catalase solution, which abolishes H_2_O_2_ activity, while the “non-peroxide” activity remains [8,18].

Honey is a promising candidate for the prolonged, topical treatment of superficial infections like tinea due to its broad-spectrum antimicrobial properties and its low toxicity. Previous studies have shown that unpasteurized H_2_O_2_-producing honeys from New Zealand and Iran were able to inhibit a range of dermatophyte species [19,20]. Australia has diverse and unique native flora, and numerous Australian honeys exhibit high H_2_O_2_-based antibacterial activity, especially those of *Eucalyptus marginata* (jarrah) origin [21].

This study set out to characterize the antifungal activity of selected Australian honeys against dermatophytes, with a focus on jarrah honey. We show here that jarrah honey was potently antifungal, inhibiting conidial germination and damaging hyphal structures. Unexpectedly, oxidative stress was not observed in the treated hyphae, and the estimated level of H_2_O_2_ produced in diluted honey was insufficient to account for the level of antifungal activity that was observed.

## 2. Results

### 2.1. Minimum Inhibitory Concentrations (MICs) and Minimum Antifungal Concentrations (MFCs) for Dermatophytes Treated with Jarrah, Leptospermum and Artificial Honey

MIC testing was undertaken for Jarrah 2017, *Leptospermum* and artificial honey, with the antifungal voriconazole (VOR) used as a control for the testing protocol. Jarrah 2017 and *Leptospermum* honey had broad antifungal activity against the collection of dermatophyte species, which was not seen when artificial honey, which replicates the high level of sugar found in honey, was used (Table 1). Jarrah 2017 honey produced the lowest MICs and MFCs, suggesting dermatophytes are particularly susceptible to peroxide activity.

When tested with additional Australian H_2_O_2_-producing honeys, the dermatophyte species *Nannizzia gypsea*, *Trichophyton interdigitale,* and *Trichophyton rubrum* were all very susceptible with MICs of 1.56–4.2% (*w*/*v*) honey (Table 2). The addition of catalase (7000 U/mL) to honey dilutions substantially increased the MICs for these species across different peroxide-types honeys, although for *T. rubrum* and *N. gypsea* an MIC of 25% (*w*/*v*) was still achievable (Table 2). This suggests that H_2_O_2_ production is critical for the potent antifungal activity of these honeys.

### 2.2. Estimation of H_2_O_2_ Production by Jarrah (Barnes 10+) Honey

Jarrah (Barnes 10+) honey had a high level of antifungal activity (Table 2) but relatively low levels of H_2_O_2_ production ([22]; Table 3). This honey was chosen for extended testing as its low H_2_O_2_ levels but high activity meant we could analyze the mode of activity of Jarrah honey without excess damage due to H_2_O_2_ production. Based on our estimate that 25% (*w*/*v*) Jarrah (Barnes 10+) honey produces ~448 μM of H_2_O_2_ at 1 h ([22]) we calculated the concentration of H_2_O_2_ present in 1× MIC for *T. rubrum* (1.56% [*w*/*v*]) to be ~28 μM (Appendix A).

### 2.3. Microscopic Examination of the Effect of Jarrah Honey on T. rubrum Conidia and Their Germination

In order to explore the effect of jarrah honey and the production of H_2_O_2_ on *T. rubrum*, we created a series of osmolarity-balanced honey solutions for Jarrah (Barnes 10+) from ½–4× MIC with artificial honey added to a final concentration of 10% (*w*/*v*) (Table 4). Similarly, we made a series of Synthetic H_2_O_2_ Honey solutions from ½–4× MIC for pure H_2_O_2_ (where the MIC = 340 µM), a second series of 0, 25, 50, 100, 200, and 400 µM H_2_O_2_ to test lower H_2_O_2_ levels, and a solution containing 56 µM H_2_O_2_ (which we calculated to be present in 2× MIC Jarrah [Barnes 10+] honey; Appendix A). These were also adjusted to an osmolarity of 10% (*w*/*v*) honey using artificial honey. A final set of Jarrah (Barnes 10+) + Catalase + Artificial honeys were created where catalase (7000 U/mL) was added to Jarrah (Barnes 10+) honey and adjusted to 10% (*w*/*v*) honey using artificial honey. The figures in the results where these different honey solutions were used are listed in Table 4.

Calcofluor white staining of the fungal cell wall was used to investigate whether H_2_O_2_ and honey affected conidia and their germination. *T. rubrum* conidia treated for 48 h with Jarrah (Barnes 10+) honey (containing artificial honey to standardize osmolarity; Table 4) or Synthetic H_2_O_2_ Honey at ½, 1, 2 and 4× MIC had very limited or no germination and were 3–7 µM long, while untreated conidia produced long germ tubes of up to 60 µM (Figure 1a; left and middle panels).

As the MIC for Synthetic H_2_O_2_ Honey (430 µM; Table 4) was substantially higher than the concentration that was calculated to be present in 1× MIC for Jarrah (Barnes 10+) honey (~28 µM), a range of concentrations that encompassed the latter were tested for their ability to inhibit conidial germination. Synthetic H_2_O_2_ Honey concentrations of 25 µM, 50 µM, and 100 µM were unable to inhibit germination compared to the untreated sample (*p* > 0.42 for all comparisons) (Figure 1a; right panel), and inhibition was only achieved at concentrations of 200 µM and 400 µM (*p* < 0.001). Representative images showing the effect of Jarrah (Barnes 10+) and Synthetic H_2_O_2_ Honey on conidial germination are shown in Figure 1b.

### 2.4. Analysis of Oxidative Stress in Fungal Hyphae Following Treatment with Jarrah (Barnes 10+) Using DCFDA and CellROX Green

Catalase treatment substantially increased the MICs of the peroxide honeys suggesting that their antifungal action is dependent on H_2_O_2_ (Table 2). Levels of H_2_O_2_ and other intracellular reactive oxygen species (ROS) that exceed cellular antioxidant defences can result in temporary or permanent oxidative changes to cellular structures including lipids, proteins and DNA [23,24]. To test whether intracellular ROS was generated from honey treatment we used two fluorophores sensitive to oxidative species: DCFDA and CellROX Green. Six and 14.5 h were chosen for treatment as most H_2_O_2_ production occurs within the first 1–6 h of honey dilution, and at 14.5 h any gradual build-up of oxidative stress should become apparent [25].

Representative images of stained hyphae are shown in Figure 2a. At 6 h, no DCFDA fluorescence was observed in the untreated control, while fluorescence was observed in the 4× MIC Synthetic H_2_O_2_ Honey treatment. None of the Jarrah (Barnes 10+) treatments, ranging from ½× MIC to 4× MIC, showed any DCFDA fluorescence.

A longer treatment time of 14.5 h was then employed to detect any accumulation of oxidative stress, and this was detected using DCFDA (Figure 2b,d) and CellROX Green (Figure 2c,e). The Synthetic H_2_O_2_ Honey control was reduced to ½× MIC (215 µM) to prevent hyphal death that might result in increased oxidative stress. Unlike the 6-h treatment, at 14.5 h the hyphae treated with 4× MIC Jarrah (Barnes 10+) honey had significantly more DCFDA fluorescence than the untreated control (Figure 2b,d; *p* < 0.001) indicating that the honey was not inhibiting DCFDA fluorescence. There was no difference in DCFDA fluorescence between the other honey treatment concentrations and the untreated sample (*p* ≥ 0.5 for all comparisons). Hyphae treated with ½× MIC Synthetic H_2_O_2_ Honey had substantially more DCFDA fluorescence than the untreated sample (Figure 2d; *p* < 0.001).

CellROX Green binds irreversibly to DNA so cannot be lost from the hyphae, unlike DCFDA, which is membrane permeable. After 14.5 h of treatment, there was significantly more CellROX Green fluorescence in the ½× MIC Synthetic H_2_O_2_ Honey (215 µM) treatment than in the untreated control (*p* < 0.001) (Figure 2c,e). Synthetic H_2_O_2_ Honey containing 56 µM H_2_O_2_ was also tested, as this is the concentration estimated to be present in 2× MIC Jarrah (Barnes 10+) honey ([22]; Appendix A). The sub-inhibitory concentrations of H_2_O_2_ caused significantly more oxidative stress in hyphae compared to the untreated sample (*p* < 0.0001). There was no difference in CellROX fluorescence between the untreated, ½× MIC and 1× MIC Jarrah (Barnes 10+) honey treatments, and the 2× MIC Jarrah (Barnes 10+) honey treatment had less fluorescence than the other treatments (Figure 2e).

### 2.5. Analysis of T. rubrum Hyphae Treated with Jarrah Honey by Scanning Electron Microscopy

Scanning electron microscopy was used to determine whether treatment with Jarrah (Barnes 10+) honey (with or without catalase; Table 4) or Synthetic H_2_O_2_ Honey causes structural changes to *T. rubrum* hyphae. Hyphae treated with artificial honey containing catalase exhibited smooth, uniform surfaces (Figure 3). In comparison, treatment with Jarrah (Barnes 10+) honey at 1× MIC resulted in prominent bulges and protrusions, although the hyphal surface remained smooth. With 2× MIC Jarrah (Barnes 10+) honey, hyphae became shrunken and collapsed and their surface appeared rough and damaged.

With the addition of catalase, Jarrah (Barnes 10+) honey at 1× and 2× MIC caused hyphae to bulge with a similar appearance to those treated with Jarrah (Barnes 10+) honey at 1× MIC. There was no evidence of the collapsing seen with 2× MIC Jarrah (Barnes 10+) honey without catalase (Figure 3). When treated with Synthetic H_2_O_2_ Honey containing 56 µM H_2_O_2,_ hyphae developed rough surfaces and some collapsing was seen. Some bulging was also present, but this was at a lower frequency than what was seen with Jarrah (Barnes 10+) honey and honey plus catalase treatments. These hyphal deformities suggest that low concentrations of H_2_O_2_ damage the surface of mature *T. rubrum* hyphae and that other stressors may be present in Jarrah (Barnes 10+) honey that cause the ballooning morphology.

## 3. Discussion

Tineas are the most common fungal infections globally, and their incidence is rising [1,2,3]. The limitations of current systemic and topical antifungal treatments have led to renewed interest in natural alternatives, including honey. Certain unique floral honeys from Australia have particularly high antimicrobial activity that is mediated by H_2_O_2_ and could be therapeutically useful for fungal infections [21]. This study aimed to characterize the activity of Australian jarrah honey against dermatophyte fungi.

### 3.1. Many Fungal Dermatophyte Species Are Highly Susceptible to Jarrah Honey

The H_2_O_2_-producing natural honeys tested in this study exhibited high antifungal activity against the six dermatophyte species (Table 1 and Table 2). Fungi are generally more susceptible to the action of H_2_O_2_-type honeys than to *Leptospermum* (manuka) honey, where activity is dependent on MGO [8,19], and filamentous fungi such as *Aspergillus, Penicillium, Microsporum* [26], and *Trichophyton* [27] have been found to have greater sensitivity to honey than *Candida* and *Saccharomyces* yeasts [26,27]. In the limited studies that have included them, dermatophytes appear to be particularly susceptible to honey activity [20,26]; and compared to our previous work on *Candida* species that also tested jarrah honey the MICs for the dermatophytes tested here are ~10-fold lower [8].

The dermatophytes appeared substantially more susceptible to jarrah honey than has been reported for other H_2_O_2_-producing honeys, where MICs have ranged from 5–39% (*v*/*v*) [19,20]. Our results are in contrast to a recent study of a commercial jarrah honey, however, which had no activity against *Trichophyton mentagrophytes* and *T. rubrum* [28]. H_2_O_2_-producing honeys can be highly variable in activity, which can be influenced by age, processing and storage conditions as well as bee health and geographic factors, and even fresh and unprocessed samples can range from very high to negligible activity [21,29]. However, if stored correctly, highly active jarrah honey samples can retain therapeutically useful activity for many years [22]. Given this apparent hyper-susceptibility to jarrah honey and the fact that dermatophytes mostly cause topical infections, our findings suggest that superficial tineas could be treated effectively using ointments or gels that incorporate active jarrah honey.

### 3.2. H_2_O_2_ Production Is Necessary but Not Sufficient for the Inhibition of Dermatophyte Fungi by Jarrah Honey

The addition of catalase dramatically reduced the activity of Jarrah (Barnes 10+) honey, indicating that H_2_O_2_ production is critical for antifungal action (Table 2). However, the level of antifungal activity of the various honey samples did not correlate with their level of H_2_O_2_ production, and the amount of H_2_O_2_ estimated to be present in Jarrah (Barnes 10+) at the MIC was ~15-fold lower than the level of H_2_O_2_ needed to inhibit *T. rubrum*, even when taking inhibition of the HRP assay into account (Appendix A; Figure 1a). Furthermore, unlike whole Jarrah (Barnes 10+) honey, artificial honey spiked with H_2_O_2_ at a similar level did not inhibit the germination of conidia (Figure 1a,b). This suggests that H_2_O_2_ alone is not sufficient for inhibition and other factors influence the antifungal effect of jarrah honey.

Studies of the antibacterial effect of honey have similarly found the concentration of hydroxyl radicals to be much lower than what is expected based on their MIC values [30]. Potential synergistic compounds that might be present in Jarrah (Barnes 10+) include polyphenols [31,32], antimicrobial peptides [33], Maillard reaction products [34], and gluconic acid [35]. Contamination of honey with antibiotics and antifungals is also a possibility, however, jarrah grows in native forests that are not subjected to agricultural sprays, and as jarrah is a high-value product, beekeepers ensure they place their hives away from other forage. In addition, Australian honey is produced under strict guidelines for chemical residues and subjected to rigorous surveillance testing, and compliance is very high. Fungicides would also persist with the addition of catalase, however, almost all the antifungal activity was lost when the honey samples were treated with catalase (Table 2). Alternatively, the inhibition of the HRP/*o*-dianisidine assay may have been greater than we calculated, causing the level of H_2_O_2_ in Jarrah (Barnes 10+) honey to be underestimated, which has been noted in other studies [21,24,32].

### 3.3. The Antifungal Activity of Jarrah Honey Appears to Be Mediated on the Surface of T. rubrum Hyphae, Causing Deformities and Hyphal Collapse

Scanning electron microscopy of mature *T. rubrum* hyphae following honey treatment revealed regions with visible swelling, bulging, and collapse (Figure 3). This damage to the hyphal surface increased with honey concentration and did not appear to be due to H_2_O_2_ as it also occurred in the presence of catalase (Figure 3). Severe hyphal collapse was apparent following treatment with 2× MIC Jarrah (Barnes 10+) honey, while treatment with artificial honey containing 56 µM H_2_O_2_ (estimated to be present in 2× MIC of Jarrah [Barnes 10+] honey) caused only minor hyphal collapse and surface roughening. As above, these findings suggest that H_2_O_2_ production in Jarrah (Barnes 10+) honey is necessary but not sufficient for its high antifungal activity against *T. rubrum*, and that there are other synergizing agents present.

The observed hyphal changes in *T. rubrum* are consistent with observations from other antifungal studies that have noted hyphal bulging leading to severe mycelial collapse [36,37,38]. The authors speculated that the cell membrane or wall becomes weakened due to direct damage, causing the structures to bulge and eventually, with lethal treatment, collapse [36,37,38]. In bacteria, treatment with either manuka or H_2_O_2_-producing honey produces changes to the surface structure including blebs and furrows, along with an increase in cell size [39,40]. This damage could be due to antimicrobial peptides present in honey, for example bee-defensin 1 [33], as these are known to elicit their cytotoxic effects through destruction of the cell membrane resulting in visible surface roughening, cellular leakage and regions of collapse [41,42].

Surprisingly, fluorescent redox dyes were unable to detect internal oxidative stress in mature hyphae following treatment with inhibitory and sub-inhibitory concentrations of honey (Figure 2). Antifungal agents that damage and kill cells normally activate stress response pathways [43,44,45], however, stress pathway down-regulation has been observed following synergistic treatments that seem to disrupt the normal stress response [46]. Yeast cells treated with propolis (a resinous substance produced by bees and containing beeswax and plant matter) show increased oxidative stress after 5–10 min [47]. In cancer cells, however, no oxidative stress was observed following honey treatment suggesting that cellular apoptosis occurred through a ROS-independent cellular pathway [48]. Together, these results suggest that the antifungal mechanism of honey is largely mediated at the hyphal surface, which is rapidly damaged without invoking intracellular ROS.

## 4. Conclusions

Our study indicates that jarrah honey has unique antifungal attributes that work to inhibit and kill dermatophytes, making it a potentially promising candidate for the treatment of tineas. Remarkably low concentrations of jarrah honey were able to inhibit conidial germination and cause significant damage to mature hyphae in vitro. Although H_2_O_2_ production was critical for antifungal activity, the level present in Jarrah (Barnes 10+) honey was too low to account for its potent activity toward dermatophytes, suggesting it may possess agents that augment the killing caused by peroxide damage. Further research aimed at fractionating jarrah honey samples could identify these potential synergists and enable rapid screening of honey samples for high activity. In vivo studies are now needed to determine whether this potent antifungal activity of jarrah honey translates to mycological cure in clinical settings.

## 5. Materials and Methods

### 5.1. Dermatophyte Cultures

Dermatophyte species isolated from clinical specimens were sourced from the collection at RMIT University, Melbourne. Isolates were sub-cultured onto potato dextrose agar (PDA) plates from PDA slants and were incubated at 30 °C until sufficient conidiation was observed. Conidia were collected by washing colonies with 1 mL sterile phosphate buffered saline (PBS) while gently abrading the surface with the tip of a pipette. The PBS was then recovered and placed in microcentrifuge tube for 10 min to allow particulate matter and hyphal fragments to settle. The upper 600 µL was then decanted into a fresh microcentrifuge tube. The concentration of conidia in the suspension was determined using a hemocytometer, and standardized conidia concentrations were prepared in RPMI-1640 (Sigma Aldrich, St. Louis, MI, USA, Cat. No. R6504) for antifungal testing, or PBS for other tests.

### 5.2. Honey Samples

Six honey samples were tested for antifungal activity (Table 3). Three samples were collected during an Australian honey survey [21] where they were found to have antimicrobial activity and moderate-to-high levels of peroxide-based activity. One sample was from the University of Technology Sydney honey collection and was derived from *Leptospermum speciosum*, with non-peroxide-based activity that is assumed to be due to MGO. Two honeys were sourced in larger volumes for extended testing: Jarrah (Barnes 10+), a commercially available edible honey with relatively low peroxide activity (marketed as Barnes Naturals Active Jarrah Honey) and purchased in 2019 [22], and Jarrah 2017, a raw jarrah honey sample collected in Northcliff WA Australia in 2017 with high peroxide activity. All honey samples were stored in the dark at 4 °C. Prior to testing, the honey samples were diluted to 50% (*w*/*v*) in H_2_O and incubated at 35 °C with shaking at 180 rpm for 20 min to aid mixing. Then were then vortexed and filter sterilized using a 0.22 μm syringe filter (Millex, Duluth, GA, USA, Cat. No. SLG033) and were used on the day of testing.

### 5.3. Quantification of H_2_O_2_ Production

H_2_O_2_ produced by the honey samples were previously quantified [22] using a recently described, standardized horseradish peroxidase (HRP)/*o*-dianisidine colorimetric assay [25]. The assay was conducted over a 24 h period with the maximum level of production recorded (Table 3); data are the mean ± SEM of two biological replicates.

In Jarrah (Barnes 10+) honey, our previous study found an average maximum of 136 µM at 1 h post dilution that diminished over time [22]. When spiked with H_2_O_2_ to give a final concentration of 500 µM, only 188 µM could be recovered after 2 h of incubation, suggesting either the HRP reaction had been inhibited or the H_2_O_2_ was being degraded during incubation. Considering this potential masking or loss of 312 µM of H_2_O_2_ production, we estimated 25% (*w*/*v*) Jarrah (Barnes 10+) honey to produce ~448 µM of H_2_O_2_ at 1 h [22]. This estimate was used to approximate the amount of H_2_O_2_ present in minimum inhibitory concentration levels of Jarrah (Barnes 10+) honey.

### 5.4. Honey Component Solutions

Various components of honey were tested by creating a series of honey component solutions (Table 4). Artificial honey, which was used as an osmotic control, was made by dissolving 2.29 g/mL fructose, 2.07 g/mL glucose, and 0.16 g/mL sucrose in dH_2_O and incubating at 40 °C for several days until the sugars were fully dissolved it became homogenous in consistency. It was then diluted to 50% (*w*/*v*) with dH_2_O, incubated at 35 °C with shaking and filter sterilized as above. This was used to balance the level of sugar in diluted samples of the Jarrah (Barnes 10+) honey to be consistently equal to 10% honey (referred to in Table 4 as “Jarrah (Barnes 10+) + artificial honey”). A series of Synthetic H_2_O_2_ Honey solutions were also prepared by adding together the 50% (*w*/*v*) artificial honey solution, 2× RPMI-1640 (Sigma Aldrich, St. Louis, MI, USA, Cat. No. R6504) and 3% H_2_O_2_ (Sigma Aldrich, Cat. No. 88597), to achieve a final concentration of 10% (*w*/*v*) artificial honey and a range of concentrations of H_2_O_2_ based on the MIC for *T. rubrum* (Table 4). All the honey solutions were freshly prepared on the day of the experiments where they were used.

### 5.5. Minimum Inhibitory Concentrations (MICs) and Minimum Fungicidal Concentrations (MFCs) of Honey

Susceptibility testing was conducted in accordance with the Clinical and Laboratory Standards Institute (CLSI) [49]. Honey and H_2_O_2_ solutions were made freshly on the day of testing. All tests containing honey or H_2_O_2_ were conducted in subdued lighting.

Minimum inhibitory concentrations (MIC) were determined for the different honey samples, H_2_O_2_ in solution (Sigma Aldrich, St. Louis, MI, USA, Cat. No. 88597), and artificial honey, with the various dermatophyte species (Table 1). Ninety-six-well, flat-bottomed plates were used for all antifungal testing. Conidial suspensions of each dermatophyte species were prepared in RPMI-1640 to a concentration of 1–3 × 10^4^ conidia/mL. Two-fold serial dilutions of the different honey or H_2_O_2_ solutions were prepared in RPMI-1640 to give final concentrations of 0.25–5% (*w*/*v*) Jarrah 2017 honey, 0.25–50% (*w*/*v*) *Leptospermum* and artificial honey, and 0.002–2.00 mM H_2_O_2_. Twenty microlitres of the conidial suspension was added to 180 µL of each honey or H_2_O_2_ dilution to give a final concentration of 1–3 × 10^3^ conidia/mL. Voriconazole (VOR; Sigma Aldrich, St. Louis, MI, USA, Cat. No. D8418) was used as a control antifungal agent and tested alongside the honey samples. A stock solution of 400 µg/mL VOR was prepared in DMSO and diluted in RPMI-1640 to give final 2-fold dilutions from 0.008–4 µg/mL. Conidia suspensions were added as above to a final concentration of 1–3 × 10^3^ conidia/mL. To test the effect of H_2_O_2_ on activity in the honey samples where peroxide activity was detected (Table 3), honey samples were serially diluted to final concentrations of 0.05–25% (*w*/*v*) as above and a parallel MIC was performed where the RPMI-1640 medium contained 7000 U/mL catalase (Sigma Aldrich, St. Louis, MI, USA, Cat. No. SRE0041). The plates were incubated for 96 h at 30 °C under humidified conditions. Positive (no inhibitory agent) and negative (no fungi) controls were included on each plate.

The lowest concentration of the antimicrobial agent to inhibit growth was defined as the minimum inhibitory concentration (MIC), with MIC_100_ and MIC_80_ indicating no growth or an 80% reduction in growth, respectively. MICs were determined visually by comparison with the positive and negative growth controls. The minimum fungicidal concentration (MFC) was determined by spotting 50 µL aliquots from wells with no apparent growth onto SDA plates and incubating these at 30 °C for 96 h. The lowest honey concentration producing less than three colonies, which corresponds to killing ≥ 99.9% of the inoculum, was defined as the MFC. MICs and MFCs were determined as the mean ± SEM (if variation was present) of two biological replicates for Table 1 and three biological replicates for Table 2.

### 5.6. Microscopic Analysis of T. rubrum Conidia Treated with Honey

*Trichophyton rubrum* was chosen for further investigation as it is the most commonly isolated dermatophyte in superficial infections globally [50]. Filter-sterilized solutions of 50% (*w*/*v*) Jarrah (Barnes 10 +) honey or artificial honey were prepared in dH_2_O and then added to 2× RPMI-1640 to create the dilutions series of Jarrah (Barnes 10+) + artificial honey and Synthetic H_2_O_2_ Honey solutions based on the MICs for *T. rubrum* (Table 4). To determine if the low levels of H_2_O_2_ predicted to be produced in Jarrah (Barnes 10+) honey were able to inhibit conidia, Synthetic H_2_O_2_ Honey solutions containing 0, 25, 50, 100, 200 and 400 µM H_2_O_2_ were also created (Table 4).

To assess how honey affected the germination of *T. rubrum* conidia, sterile square (22 × 22 mm) microscope coverslips (Trajan Scientific and Medical, Ringwood, Australia, Cat. No. 471112222) were placed into each well of a six-well flat-bottomed plate (Corning Costar, Corning, AZ, USA, Cat. No. 3516) and covered with 1.5 mL of the appropriate treatment solution. Each well was then inoculated with 500 μL of 8 × 10^4^ conidia/mL *T. rubrum* conidia in PBS (prepared as outlined in Section 5.1) to give a final concentration of 2 × 10^4^ conidia/mL. The plates were incubated at 30 °C for 48 h. Following this, coverslips were rinsed in PBS, then stained with 1 mL of 50 µM calcofluor white (Sigma Aldrich, St. Louis, MI, USA, Cat. No. 18909) in PBS and incubated for 1 min protected from light.

Conidia and germinating hyphae were visualized using the DAPI filter set of an Olympus BX51 microscope (Shinjuku, Japan), and images were taken using the 40× objective. The length of conidia and hyphae were measured using Fiji Image J software [51], where the straight-line tool connected the ends of structures.

### 5.7. Detection of Reactive Oxygen Species in T. rubrum Hyphae Treated with Jarrah Honey

A similar method to that outlined in Section 5.6 was used to detect oxidative stress in *T. rubrum* hyphae treated with honey. A square sterile (22 × 22 mm) coverslip was placed into each well of a 6-well flat-bottomed plate, which was then inoculated with 2 × 10^4^ conidia/mL in 3 mL RPMI-1640. The plates were incubated at 30 °C for 24 h to generate hyphae, and the media was then removed and replaced with Jarrah (Barnes 10+) + artificial honey at ½× MIC, 1× MIC, 2× MIC and 4× MIC (Table 4). Synthetic H_2_O_2_ Honey was used as a positive control for ROS production, with H_2_O_2_ at 56 µM (equivalent to 2× MIC in Jarrah (Barnes 10+) honey; see Section 5.3), ½× MIC (215 µM) or 4× MIC (1720 µM; Table 4). The hyphal cultures were incubated at 30 °C for 6 or 14.5 h to detect both early and final ROS production. Following incubation, the media was removed by aspiration and coverslips with adherent hyphae were rinsed once in PBS and then incubated in 1 mL RPMI-1640 medium containing 50 µM calcofluor white and 10 µM 2′,7′-dichlorofluorescin diacetate (DCFDA) (Sigma-Aldrich, St. Louis, MI, USA, Cat. No. D6883), or 50 µM calcofluor white and 5 µM CellROX Green (Sigma-Aldrich, St. Louis, MI, USA, Cat. No. C10444), for 30 min at 30 °C with shaking at 45 rpm. The stains were removed by aspiration and coverslips were then rinsed three times with PBS and imaged using an Olympus BX51 microscope equipped with a DAPI or FITC filter. Exposure was kept consistent at 200 ms. To calculate the fluorescence intensity of DCFDA staining, each calcofluor white-stained hyphal structure seen under the DAPI filter was outlined in Image J and the area measured. This hyphal outline was then copied onto the image of the hypha taken under the FITC filter and the integrated density of fluorescence from the outlined area was measured. Three background readings were taken around the perimeter of the hypha in the FITC image. The corrected total cell fluorescence (CTCF) for DCFDA was determined using the following equation:CTCF *=* integrated density *−* (area of hyphae *×* mean fluorescence of background)

### 5.8. Scanning Electron Microscopy of T. rubrum Hyphae Treated with Jarrah Honey

Scanning electron microscopy (SEM) was used to reveal hyphal changes in greater detail. Round Thermanox coverslips (13 mm diameter) (ProSciTech, Thuringowa, Australia, Cat. No. GL083) were first sterilized by washing in 90% acetone then Milli-Q H_2_O and were then incubated in 1% polyethyleneimine (Sigma Aldrich, St. Louis, MI, USA, Cat. No. 03880) and rinsed twice in Milli-Q H_2_O. Slides were left to dry at room temperature for 3 h. Treated coverslips were then placed into six wells of a 24-well flat-bottomed plate that each contained 500 µL RPMI-1640 medium containing 4 × 10^4^
*T. rubrum* conidia/ml. Plates were incubated for 24 h at 30 °C, after which the media was removed and replaced with 500 µL of each of the following treatment solutions: 1× MIC Jarrah (Barnes 10+) + artificial honey ± 7000 U catalase; 2× MIC Jarrah (Barnes 10+) + artificial honey ± 7000 U catalase; Synthetic H_2_O_2_ Honey containing 56 µM H_2_O_2_ (predicted to be present in 2× MIC of Jarrah [Barnes 10+] honey); for all treatments osmolarity was standardized to a final concentration of 10% (*w*/*v*) honey (Table 4). A control treatment contained 500 µL 10% (*w*/*v*) artificial honey + 7000 U catalase. Following a 24-h incubation at 30 °C, treatment solutions were removed by aspiration and coverslips were then washed in 0.1 M phosphate buffer (PB). Slides were then fixed by adding 250 µL of primary fixative containing 2.5% glutaraldehyde (Chem-Supply, Gillman, Australia, Cat. No. EA043) and 2% paraformaldehyde (Sigma Aldrich, St. Louis, MI, USA, Cat. No. 47608) in 0.1 M PB to each well, with overnight incubation at 4 °C.

The primary fixative solution was removed by aspiration and coverslips were then rinsed three times for 5-min with 500 µL of 0.1 M PB. Rinsed coverslips were incubated in ~250 µL (enough to completely submerge coverslip) of secondary fixative containing 2% (*v*/*v*) OsO_4_ (ProSciTech, Thuringowa, Australia, Cat. No. C011) in 0.1 M PB, for 1 h. This was removed and followed with three × 5-min washes with 250 µL of Milli-Q H_2_O. An ethanol concentration gradient was used to dehydrate fixed samples consisting of two treatments of 30, 50 and 70% (*v*/*v*) ethanol, then three treatments of 95 and 100% (*v*/*v*) ethanol. Further chemical desiccation of samples was conducted by adding 250 µL hexamethyldisilazane (HMDS) (Sigma Aldrich, St. Louis, MI, USA, Cat. No. 440191) for 2 min. HMDS was removed and samples were allowed to dry in the fume hood. Coverslips containing desiccated hyphae were mounted onto metallic stubs and an Emitech K550X sputter coater (Paris, France) covered them in gold at 2 mA for 2 min. Samples were visualized under 15 kV using a JEOL JCM-6000 benchtop scanning electron microscope (Tokyo, Japan). 

### 5.9. Statistical Analysis

Hyphal length measurements and fluorescence intensity data from experiments 4.6 and 4.7, respectively, were analyzed in GraphPad Prism (San Diego, CA, USA). The D’Agostino-Pearson omnibus normality test was conducted on hyphal germination and fluorescence data sets to determine whether the data exhibited a Gaussian distribution. As all data had a non-Gaussian distribution, they were analyzed using the non-parametric Kruskal–Wallis test. Significant results were followed up with a Dunn’s multiple comparison test that determined the treatment groups that were significantly different to each other. In all figures, asterisks denote the following: * *p* ≤ 0.05 ** *p* ≤ 0.01, *** *p* ≤ 0.001, **** *p* ≤ 0.0001.

## Figures and Tables

**Figure 1 pathogens-10-00194-f001:**
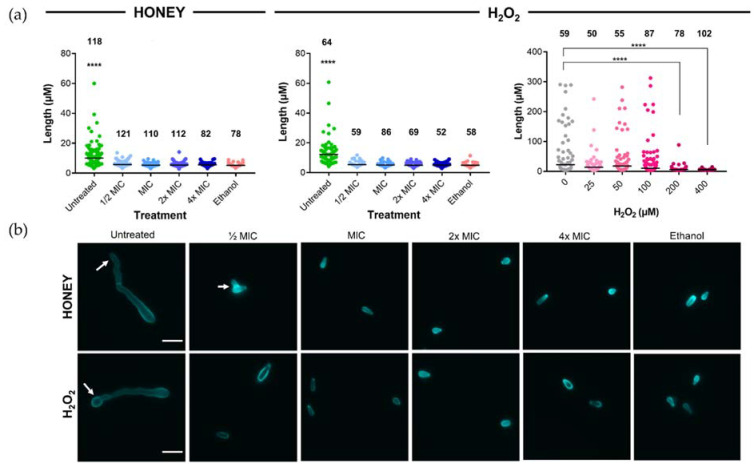
Jarrah (Barnes 10+) honey and Synthetic H_2_O_2_ Honey inhibit the germination of *T. rubrum* conidia. (**a**) Length of hyphae and conidia following treatment for 48 h with ½× MIC–4× MIC of Jarrah (Barnes 10+) honey (**left**), ½× MIC–4× MIC of Synthetic H_2_O_2_ Honey (**middle**) or Synthetic H_2_O_2_ Honey starting at around the level estimated to be present in Jarrah (Barnes 10+) honey (**right**). ½× MIC levels of Jarrah (Barnes 10+) and ½× MIC Synthetic H_2_O_2_ Honey completely suppressed conidial germination, however, Synthetic H_2_O_2_ Honey that contained a level of H_2_O_2_ calculated to be close to that present in 1× MIC Jarrah (Barnes 10+) honey (25 µM) was not able to do this, with suppression seen only when the concentration was ≥200 µM. Ethanol at 70% (*v*/*v*) was used as a positive control. Bars represent the median and numbers above the data indicate the number of hyphae/conidia measured. Data in the left and middle panels are representative of two independent replicates. (**b**) Representative images of conidia and hyphae stained with calcofluor white following treatment with Jarrah (Barnes 10+) honey (**top** row) or Synthetic H_2_O_2_ Honey (**bottom** row). Arrows indicate germinated conidia. The scale bar represents 10 µm. **** *p* ≤ 0.0001.

**Figure 2 pathogens-10-00194-f002:**
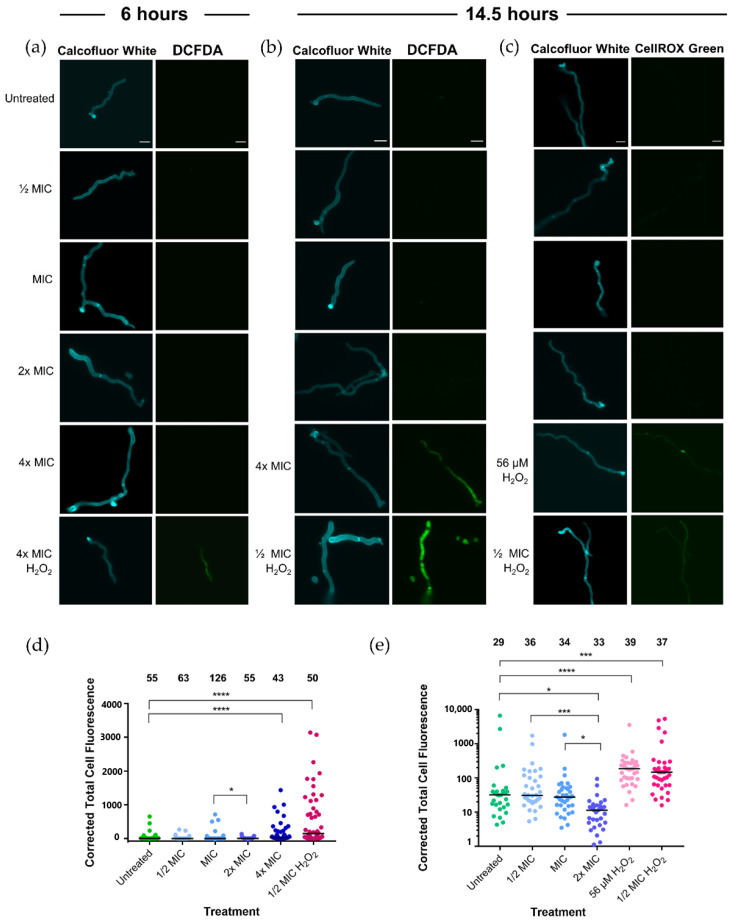
Inhibitory concentrations of Jarrah (Barnes 10+) honey do not induce intracellular oxidative stress. Representative images of calcofluor white fluorescence and (**a**) 2′,7′-dichlorofluorescin diacetate (DCFDA) fluorescence of hyphae after 6 h, (**b**) DCFDA fluorescence after 14.5 h and (**c**) CellROX Green fluorescence after 14.5 h. Synthetic H_2_O_2_ Honey at ½× and 4× MIC were used as positive controls for intracellular oxidative stress. The scale bars represent 10 µm. Corrected total cell fluorescence of (**d**) DCFDA and (**e**) CellROX Green for hyphae treated for 14.5 h indicate significant differences between treatments. Note that for (**e**) the *Y*-axis is a log scale and therefore three values close to zero could not be graphed but were included in the statistical calculations. Bars indicate the median and numbers above the data indicate the number of hyphae measured (n). * *p* ≤ 0.05, *** *p* ≤ 0.001, **** *p* ≤ 0.0001.

**Figure 3 pathogens-10-00194-f003:**
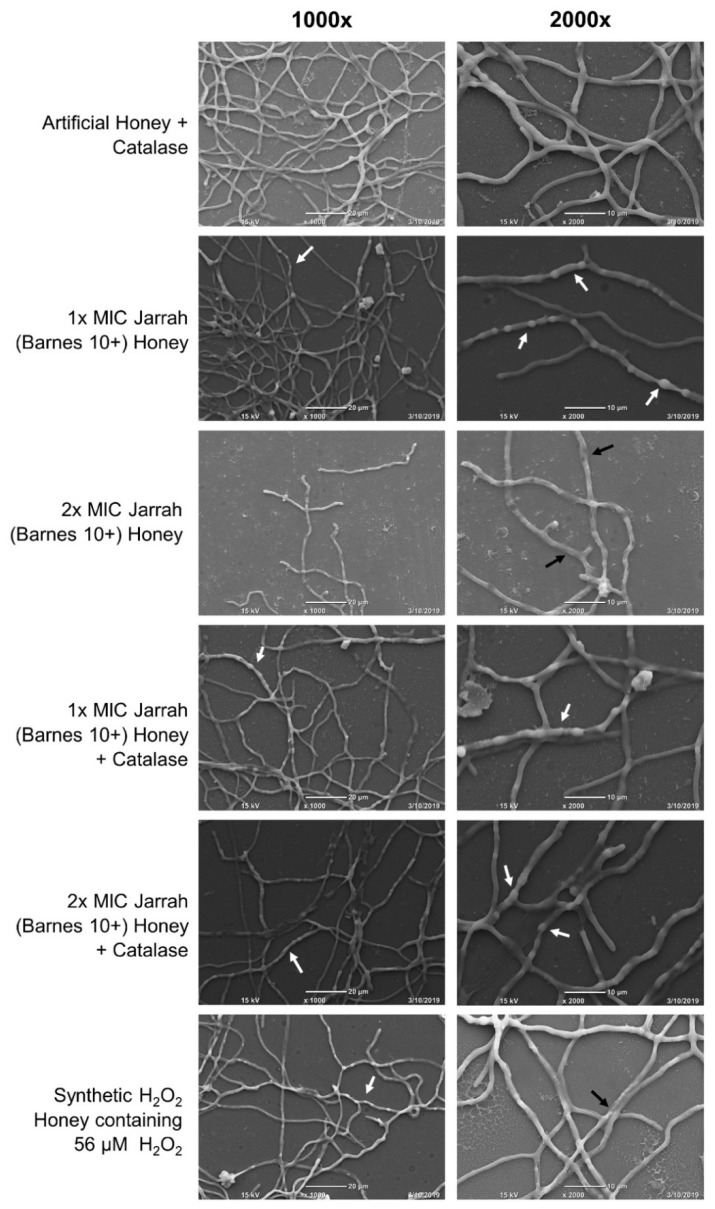
*Trichophyton rubrum* hyphae treated with Jarrah (Barnes 10+) honey have morphological deformities that are distinct from treatment with Synthetic H_2_O_2_ Honey and are not prevented by catalase. Hyphae grown for 24 h were treated with Jarrah (Barnes 10+) honey, Jarrah (Barnes 10+) honey + catalase, or Synthetic H_2_O_2_ Honey containing 56 µM H_2_O_2_ (the concentration estimated to be present in 2× MIC Jarrah [Barnes 10+] honey). Representative photos of hyphae taken at 1000× and 2000× magnification. White arrows indicate regions of hyphal ballooning and black arrows show areas of hyphal collapse.

**Table 1 pathogens-10-00194-t001:** Antifungal activity of different honey types on a collection of dermatophyte species.

Species	Honey (% (*w*/*v*))						VOR ^3^ MIC_80_ (µg/mL)
	Jarrah 2017		*Leptospermum*		Artificial		
	MIC_80_ ^1^	MFC ^2^	MIC_80_	MFC	MIC_80_	MFC	
*Microsporum canis*	1.5	2	7.5	10	20	>50	0.023
*Microsporum nanum*	1.5	2	10	10	30	>50	0.031
*Nannizzia gypsea*	3.5	5	15	25	40	>50	0.094
*Trichophyton interdigitale*	3.5	5	15	17.5	40	>50	0.125
*Trichophyton rubrum*	2.5	4	10	12.5	30	>50	0.031
*Trichophyton tonsurans*	2.5	3.5	10	10	30	>50	0.125

^1^. MIC_80_ = Minimum Inhibitory Concentration of the antimicrobial agent producing an 80% reduction in fungal growth relative to untreated control; ^2^. MFC = Minimum Fungicidal Concentration; ^3^. VOR = voriconazole.

**Table 2 pathogens-10-00194-t002:** Effect of catalase treatment on the antifungal properties of jarrah and stringybark honey.

	MIC_100_ [% (*w*/*v*)]					
	Jarrah (Barnes 10+)		Jarrah 10/13		Stringybark 19	
Catalase Treatment	−	+	−	+	−	+
*Nannizzia gypsea*	3.1	25	1.56	25	3.1	25
*Trichophyton interdigitale*	3.1	≥25	1.56	≥25	4.2 ± 1.0	>25
*Trichophyton rubrum*	1.56	25	1.56	25	1.56	25

**Table 3 pathogens-10-00194-t003:** Honey samples used in this study.

Honey Sample	Floral Source	Active Component	Maximum Hydrogen Peroxide (H_2_O_2_) Production (mM)	Origin
Jarrah 10	*Eucalyptus marginata*	H_2_O_2_	2.86 ± 0.31	[21]
Jarrah 13	*Eucalyptus marginata*	H_2_O_2_	3.84 ± 0.24	[21]
Stringybark 19	*Eucalyptus species*	H_2_O_2_	0.93 ± 0.12	[21]
*Leptospermum* 2	*Leptospermum speciosum*	MGO	–	UTS honey collection
Jarrah (Barnes 10+)	*Eucalyptus marginata*	H_2_O_2_	0.136 ± 0.01	Commercial
Jarrah 2017	*Eucalyptus marginata*	H_2_O_2_	1.59 ± 0.18	Capilano

**Table 4 pathogens-10-00194-t004:** Honey solutions created to test a range of concentrations of Jarrah (Barnes 10+) honey and H_2_O_2_ based on the MIC for *T. rubrum* while keeping osmolarity consistent at 10% (*w*/*v*) honey.

Honey Solution	Final Concentration of Components in RPMI-1640				Figures
	Artificial Honey	Jarrah (Barnes 10+) Honey	H_2_O_2_	Catalase	
	[% (*w*/*v*)]	[% (*w*/*v*)]	[µM]	+/–	
Control					
Untreated	10	0	0	–	Figure 1, Figure 2
Jarrah (Barnes 10+) + Artificial Honey					
½× MIC	9.22	0.78	0	–	Figure 1, Figure 2
1× MIC	8.44	1.56	0	–	Figure 1, Figure 2, Figure 3
2× MIC	6.88	3.12	0	–	Figure 1, Figure 2, Figure 3
4× MIC	3.76	6.24	0	–	Figure 1, Figure 2a,b,d
Synthetic H_2_O_2_ Honey					
½× MIC	10	0	215	–	Figure 1, Figure 2b–e
1× MIC	10	0	430	–	Figure 1
2× MIC	10	0	860	–	Figure 1
4× MIC	10	0	1720	–	Figure 1, Figure 2a
0 µM	10	0	0	–	Figure 1a
25 µM	10	0	25	–	Figure 1a
50 µM	10	0	50	–	Figure 1a
100 µM	10	0	100	–	Figure 1a
200 µM	10	0	200	–	Figure 1a
400 µM	10	0	400	–	Figure 1a
56 µM H_2_O_2_	10	0	56	–	Figure 2c,e, Figure 3
Jarrah (Barnes 10+) + Catalase + Artificial Honey					
Control (Artificial honey + Catalase)	10	0	0	+	Figure 3
1× MIC Jarrah (Barnes 10+) Honey + Catalase	8.44	1.56	0	+	Figure 3
2× MIC Jarrah (Barnes 10+) Honey + Catalase	6.88	3.12	0	+	Figure 3

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
