# Peer review of "Inhibition of Dermatophyte Fungi by Australian Jarrah Honey"

_pathogens, 2021, doi:10.3390/pathogens10020194_

Round 1

Reviewer 1 Report

Abstract is very poor, not clear for the reader and do not summarize the work. For instance: no introduction to the topic. Tittle should be reformulated. Add more details, emphasize the importance of the results.

The health-promoting and healing properties of honey have been known since antiquity. One of their first applications was to support the healing of wounds, which was then possible thanks to the maintenance of adequate moisture in the healing skin. Significant part of the research focuses on honeys that occur only locally in various regions the world.

The antimicrobial activity of honey depends on various factors acting both singly and synergistically. The most important ones include the content of hydrogen peroxide, phenolic compounds, and the pH of honey and osmotic pressure - the authors should mention this.

The properties of honey resulting from their chemical composition depend to a large extent from what plant species they come from. Chemical compounds contained in honey can be used in the treatment of infections of bacterial or fungal etiology, even caused by antibiotic-resistant strains of microorganisms.

Materials and methods section is incomplete and tables containing results should be improved (Headings and subtitles are unclear. State all instrumentation and chemicals/reagents used in a single subtitle. Describe analytical steps clearly for both techniques).

Graphs are illegible, please correct it.

Experimental section is documented unsatisfactorily. Headings and subtitles are unclear. State all instrumentation and chemicals/reagents used in a single subtitle. Describe analytical steps clearly for both techniques.

The statistics are unclear. Which tests were used.

The conclusions must reflect the innovation of this study and the perspectives. The results should be more emphasized to interest readers in the subject. At work, I did not have a specific summary and describe the possibilities of using the results obtained in industrial practice.

Author Response

We appreciate the detailed review provided here. Please note that the paper was already revised once on the advice of the editor after this review was written and some of the problems raised here may have already been addressed.

Abstract is very poor, not clear for the reader and do not summarize the work. For instance: no introduction to the topic. Tittle should be reformulated. Add more details, emphasize the importance of the results.

The abstract has been extensively revised with more introductory material added. We have changed the title to: Inhibition of Dermatophyte Fungi by Australian Jarrah Honey.  We hope this is now acceptable.

The antimicrobial activity of honey depends on various factors acting both singly and synergistically. The most important ones include the content of hydrogen peroxide, phenolic compounds, and the pH of honey and osmotic pressure - the authors should mention this.

This is now in the paper at abstract lines 11-12 and paragraph 3 of the introduction.

Materials and methods section is incomplete and tables containing results should be improved (Headings and subtitles are unclear. State all instrumentation and chemicals/reagents used in a single subtitle. Describe analytical steps clearly for both techniques).

We have extensively revised the materials and methods and hope that they are now satisfactory. Please note that the heading for Table 4 was removed during the submission process and appended to the results text, which has now been corrected. Hopefully this won't occur during the resubmission. 

Graphs are illegible, please correct it.

We refer the reviewer to the high-quality figures that are uploaded separately. None of the other reviewers had a problem with them.

Experimental section is documented unsatisfactorily. Headings and subtitles are unclear. State all instrumentation and chemicals/reagents used in a single subtitle. Describe analytical steps clearly for both techniques.

We have extensively revised the materials and methods to include details of the instruments and chemicals and detailed protocols. We have revised the headings and subtitles and hope that they are now satisfactory.

The statistics are unclear. Which tests were used.

We have revised the description of the statistics.

The conclusions must reflect the innovation of this study and the perspectives. The results should be more emphasized to interest readers in the subject. At work, I did not have a specific summary and describe the possibilities of using the results obtained in industrial practice.

We have extensively revised the conclusions along the lines of these recommendations.

Reviewer 2 Report

The authors examined antifungal activity of a certain Australian honey against dermatophyte fungi. The experiments are well designed. The figures are clear and illustrative. The statistical analysis is appropriate.

Author Response

The authors examined antifungal activity of a certain Australian honey against dermatophyte fungi. The experiments are well designed. The figures are clear and illustrative. The statistical analysis is appropriate.

We thank the reviewer for these supportive comments.

Reviewer 3 Report

The authors need to clearly show the production of H2O2 by honey as well as to use a well-characterized honey, such as manuka, to confirm the data.

Moreover, the ms needs an important style and language revisions.

Author Response

The authors need to clearly show the production of H2O2 by honey as well as to use a well-characterized honey, such as manuka, to confirm the data.

Hydrogen production data is presented in Table 3 for each of the honey samples. These data are taken from our previous study (Guttentag, A.; Krishnakumar, K.; Cokcetin, N.; Harry, E.; Carter, D.A. Factors affecting the production and measurement of hydrogen peroxide in honey samples. Access Microbiology 2021, https://doi.org/10.1099/acmi.0.000198).  We have amended the text to clarify this (section 4.3).

Manuka (Leptosperumum) honey was used in the first part of this study and is included in Table 1. Please note that it operates via a different mode of action and as such is not a very relevant control for this study, especially as it is much less active against dermatophytes.

Moreover, the ms needs an important style and language revisions.

We have extensively revised the manuscript and hope it is now considered suitable for publication.

Reviewer 4 Report

Abstract
into the abstract must be reported some important features strictly releated to antimicrobial activity of honey (Hydrogen peroxide, phenolic compounds, pH, sugar, ecc). Also, antimicrobial activity of honey is reported to be releated to botanical origin and it is depend to geographic area.
Please see https://www.mdpi.com/2306-7381/7/4/181
https://www.sciencedirect.com/science/article/pii/S002364381730419X

I found very confusing the titles of paragraphs in the result and discussion sections. Please make it in line with the text

No evaluation about the antibiotic presence in tested honey. Why? The results could be due to antibiotic and not releated to honey. This point is very crucial for this study.

This is my personal curiosity: why Authors used D'Agostino-Pearsono omnibus test to analyze your results?

Discussion should be revised, especially on the basis of the results viewed considering the negative test for the antibiotic residues in honey.

material and methods should be better detailed.

Author Response

Abstract
into the abstract must be reported some important features strictly releated to antimicrobial activity of honey (Hydrogen peroxide, phenolic compounds, pH, sugar, ecc).

The abstract has been extensively revised and now includes this information.

Also, antimicrobial activity of honey is reported to be releated to botanical origin and it is depend to geographic area.
Please see https://www.mdpi.com/2306-7381/7/4/181
https://www.sciencedirect.com/science/article/pii/S002364381730419X

We have cited our study of a survey of Australian honey that shows the influence of geography (Irish, J.; Blair, S.; Carter, D.A. The antibacterial activity of honey derived from Australian flora. PLoS One 2011, 6, e18229, doi:10.1371/journal.pone.0018229; line 238). This is more relevant to our paper than the papers that are cited by the reviewer.

I found very confusing the titles of paragraphs in the result and discussion sections. Please make it in line with the text

These have been extensively revised.

No evaluation about the antibiotic presence in tested honey. Why? The results could be due to antibiotic and not releated to honey. This point is very crucial for this study.

Australian honey is rigorously tested for residues and is highly compliant with guidelines. The honey samples tested were donated by beekeepers but are from batches that were destined for commercial sale, and they would have adhered closely to all guidelines. Furthemore, the honey samples are produced from native tree species in natural forests where antifungals are not used.  We can safely assume that our honey samples are not contaminated with antibiotics/antifungals. We have put a statement in the discussion to this effect (lines 261-264).

This is my personal curiosity: why Authors used D'Agostino-Pearsono omnibus test to analyze your results?

This was done to test whether the data fitted a Gaussian distribution. 

Discussion should be revised, especially on the basis of the results viewed considering the negative test for the antibiotic residues in honey.

Please see comment above.

material and methods should be better detailed.

We have extensively revised the materials and methods and hope that they are now satisfactory.

Round 2

Reviewer 1 Report

The article has been corrected. I accept the article

Author Response

The article has been corrected. I accept the article

Thank you for reviewing and accepting our article.

Reviewer 3 Report

I suggest to accept the ms

Author Response

I suggest to accept the ms.

Thank you for reviewing and accepting our paper.

Reviewer 4 Report

"Australian honey is rigorously tested for residues and is highly compliant with guidelines. The honey samples tested were donated by beekeepers but are from batches that were destined for commercial sale, and they would have adhered closely to all guidelines. Furthemore, the honey samples are produced from native tree species in natural forests where antifungals are not used.  We can safely assume that our honey samples are not contaminated with antibiotics/antifungals. We have put a statement in the discussion to this effect (lines 261-264)."

I well understand the point disccussed by Authors, but test honey to detect antibiotic residues. The control, althought rigorous, could not cover all producted honey. Moreover, honey bee during their foraging activity cover a wide area, and they could be in contact with antibiotic, i.e used in farm.

In my opion this point is very important to validate the Authors work.

Author Response

I well understand the point disccussed by Authors, but test honey to detect antibiotic residues. The control, althought rigorous, could not cover all producted honey. Moreover, honey bee during their foraging activity cover a wide area, and they could be in contact with antibiotic, i.e used in farm.

Residue testing is complex and out of the scope of our laboratory, so this is a substantial request and will not be simple to fulfill. If the reviewer could point us to a published paper where antibiotic residue testing is performed in honey (and found to be a problem, particularly in Australia) we might consider this request to be reasonable. Otherwise we consider it untenable because:

  1. To our knowledge this is not done in any antimicrobial honey papers; it has certainly never been a requirement for any of the numerous antimicrobial honey papers published by our group.
  2. Fungi are not susceptible to antibiotics, only antifungals.
  3. Australian beekeepers are very aware of the requirements for minimal residues and therefore very careful to ensure their hives are not near to areas where antifungals are used.
  4. Jarrah forests are vast and well away from agricultural lands, and bees only forage for ~2-3 km and would not be accessing other crops, particularly as beekeepers are aware of the issues of chemical contamination noted above, and also because jarrah honey is a premium product and they do not want it contaminated with nectar from other flora.
  5. If antifungal residues were present they would not be abolished by catalase, which we see happens in our honey samples (table 2).

Round 3

Reviewer 4 Report

I well understand the reasonable Authors problems.

To avoid any misunderstanding, Authors must reported in the MS the concept expressed in the response list point (1,3 and 5), better if supported by references, in order to clarify better this point 

Author Response

We have extended the discussion about fungicide residues (Section 3.2; paragraph 2):

"Contamination of honey with antibiotics and antifungals is also a possibility, however jarrah grows in native forests that are not subjected to agricultural sprays, and as jarrah is a high-value product beekeepers ensure they place their hives away from other forage.  In addition, Australian honey is produced under strict guidelines for chemical residues and subjected to rigorous surveillance testing, and compliance is very high. Fungicides would also persist with the addition of catalase, however almost all antifungal activity was lost when the honey samples were treated with catalase (Table 2). "

As these statements are based on accepted best beekeeper practice and industry guidelines there aren't any journal articles that we can reference. We hope what is written is sufficient to clear up any misunderstandings.

Round 4

Reviewer 4 Report

This version is better.